# DeepCompress: A Dual Reward Strategy for Dynamically Exploring and Compressing Reasoning Chains

**Tian Liang**[*,1]  **Wenxiang Jiao**  **Zhiwei He**[1]  **Jiahao Xu**[1]  **Haitao Mi**[1]  **Dong Yu**[1]
[1]Tencent AI Lab

 https://github.com/Skytliang/DeepCompress

## Abstract

Large Reasoning Models (LRMs) have demonstrated impressive capabilities but suffer from cognitive inefficiencies like "overthinking" simple problems and "underthinking" complex ones. While existing methods that use supervised fine-tuning (SFT) or reinforcement learning (RL) with token-length rewards can improve efficiency, they often do so at the cost of accuracy. This paper introduces **DeepCompress**, a novel framework that simultaneously enhances both the accuracy and efficiency of LRMs. We challenge the prevailing approach of consistently favoring shorter reasoning paths, showing that longer responses can contain a broader range of correct solutions for difficult problems. DeepCompress employs an adaptive length reward mechanism that dynamically classifies problems as "Simple" or "Hard" in real-time based on the model's evolving capability. It encourages shorter, more efficient reasoning for "Simple" problems while promoting longer, more exploratory thought chains for "Hard" problems. This dual-reward strategy enables the model to autonomously adjust its Chain-of-Thought (CoT) length, compressing reasoning for well-mastered problems and extending it for those it finds challenging. Experimental results on challenging mathematical benchmarks show that DeepCompress consistently outperforms baseline methods, achieving superior accuracy while significantly improving token efficiency.

## 1 Introduction

Large Reasoning Models (LRMs) have demonstrated significant advancements, exemplified by OpenAI's o1 series (OpenAI, 2024), DeepSeek's R1 (Guo et al., 2025), Google's Gemini 2.5 (Google, 2025), and Anthropic's Claude 3.7 (Anthropic, 2025). These models exhibit remarkable capabilities across diverse complex reasoning tasks. Key characteristics of LRMs include their ability to perform self-verification, engage in reflection, and generate extended Chain-of-Thought (CoT) reasoning, leading to improved accuracy. However, recent research reveals inherent inefficiencies in LRM cognition. These include *overthinking* (Chen et al., 2024), characterized by excessive intermediate step generation for simple problems, and *underthinking* (Wang et al., 2025b), manifesting as frequent, unstable thought shifts during complex problem-solving. These findings underscore the necessity for adaptive strategies to enhance both the efficiency and accuracy of current LRMs.

Recent studies have explored various strategies to enhance the reasoning efficiency of LRMs. One line of research leverages Supervised Fine-Tuning (SFT) on curated datasets of shortened CoT exemplars (Chen et al., 2024; Kang et al., 2025; Yu et al., 2025b). This approach trains LRMs to infer correct answers using fewer intermediate reasoning steps. Conversely, another line of work incorporates token-length reward functions into Reinforcement Learning (RL) frameworks (Team et al., 2025; Luo et al., 2025; Aggarwal & Welleck, 2025; Liu et al., 2025). These methods explicitly optimize for shorter reasoning paths while penalizing unnecessarily verbose ones. Although these compression techniques achieve significant efficiency gains, they are often accompanied by slight accuracy trade-offs. Therefore, the fundamental challenge remains in simultaneously achieving both superior accuracy and computational efficiency.

---

*Corresponding to *ttianliang@tencent.com*.

In this paper, we propose **DeepCompress**, a novel framework that incorporates an adaptive length reward mechanism which dynamically adjusts the preference for shorter or longer responses based on the problem difficulty perceived in real-time by the LRMs. Specifically, we first reveal that longer responses contain a wider coverage of potentially correct solutions than shorter ones for the same problems. In other words, current methods that constantly optimize shorter responses in RL processes may constrain the problem-solving capacity of LRMs and restrict their reasoning boundary. At the meantime, it is infeasible to encourage LRMs to always favor longer responses in RL processes considering the efficiency for both training and inference. Our DeepCompress addresses this challenge by first dividing the problems into "Simple" or "Hard" classes and then applying different length reward modes to them, respectively, during training. With Group Relative Policy Optimization (GRPO, Shao et al., 2024) as our basic RL algorithm, we consider a problem as "Simple" when its group pass ratio (i.e., the proportion of correct samples among its $G$ generated responses) exceeds the batch pass ratio (i.e., average of group pass ratio in the batch), and "Hard" when otherwise. Then, DeepCompress encourages the LRMs to favor shorter responses of the "Simple" problems, but longer responses of the "Hard" problems. Through this mechanism, DeepCompress dynamically adapts the reasoning chain length – autonomously compressing lengthy CoT for well-mastered problems while extending CoT for under-learned cases.

The contributions of this paper are summarized as below:

- We propose DeepCompress, which incorporates a model-aware mechanism to dynamically classify questions by difficulty, and a dual length reward to adaptively explore longer responses for "Hard" questions and favor shorter responses for "Simple" ones.

- Experimental results on challenging mathematical benchmarks demonstrate the capability of DeepCompress in achieving superior performance consistently over baseline methods while also improving the token efficiency significantly.

- Our in-depth analysis reveals that DeepCompress fosters a more effective learning process by encouraging high policy entropy. This promotes efficient exploration and reflection, leading to superior performance, particularly on challenging problems.

## 2 RELATED WORK

**Manipulating Reasoning Length through Prompt Engineering**  Research on reasoning length in LLMs presents a central trade-off. Some studies show that longer reasoning paths can improve task performance (Jin et al., 2024), whereas others advocate for conciseness to boost inference efficiency, using strategies such as Constrained-CoT (CCoT) (Nayab et al., 2024). To navigate this complexity, several methods have been proposed to optimize or adapt the reasoning process. Recognizing that excessive length in Chain-of-Thought (CoT) can impair performance, Yang et al. (2025) developed the Thinking-Optimal Scaling strategy to find an ideal length by filtering for the shortest correct reasoning paths. Other approaches focus on dynamic adaptation to the specific problem. Adaption-of-Thought (ADOT), for example, addresses the mismatch between question difficulty and prompting complexity (Xu et al., 2024), while TALE directly manages token overhead by dynamically tuning the number of reasoning tokens via the prompt (Han et al., 2024).

**Post-Training for Reasoning Efficiency**  A significant body of work improves LLM reasoning efficiency through post-training, primarily via Supervised Fine-Tuning (SFT) and Reinforcement Learning (RL). SFT-based methods train models on datasets of curated, concise reasoning exemplars, which can be generated by stronger LLMs (Chen et al., 2024; Kang et al., 2025) or used within a weighted objective that adapts the reasoning budget to question difficulty (Yu et al., 2025b). The majority of approaches, however, utilize RL to penalize excessive length. In its direct form, this involves a simple length-based reward to encourage brevity (Team et al., 2025; Luo et al., 2025; Arora & Zanette, 2025). More advanced methods employ dynamic reward-shaping, which calibrates the length penalty based on task difficulty (Cheng et al., 2025), response correctness (Yuan et al., 2025), self-supervised optimal length signals (Yi et al., 2025; Liu et al., 2025), or explicit user constraints (Aggarwal & Welleck, 2025). Innovations also extend to the training architecture itself, through methods like auxiliary reflection models (Deng et al., 2025) and iterative pruning (Hou et al., 2025). These approaches achieve notable efficiency gains, yet they offer limited accuracy improvements and occasionally incur minor performance losses.

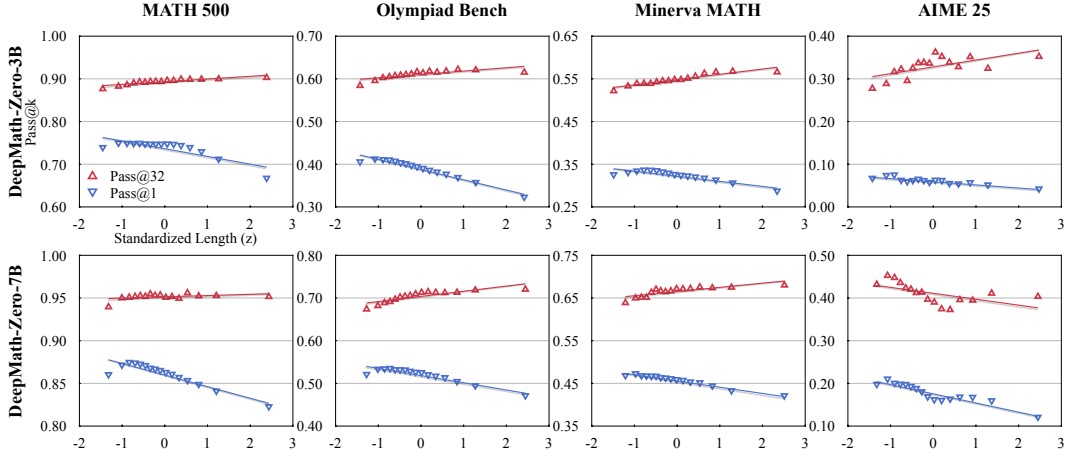

Figure 1: Relationship between standardized response length (z) and mathematical reasoning performance (pass@k). Pass@1 score decreases with increasing length, while Pass@32 generally increases.

## 3 PRELIMINARIES

Existing work has made significant efforts to reduce model response length, but this has concurrently led to a degradation in performance. In this section, we designed preliminary experiments to further analyze the relationship between response length and performance.

### 3.1 EXPERIMENTAL SETUP

**Data**   To assess the mathematical performance of our models, we followed Zeng et al. (2025) and evaluated them on four challenging benchmarks: MATH-500 (Hendrycks et al., 2021b), Olympiad-Bench (He et al., 2024), Minerva Math (Lewkowycz et al., 2022) and AIME 2025 (MAA, a).

**Model**   We conducted experiments on `DeepMath-Zero-3B`[1] and `DeepMath-Zero-7B`[2], which are created by finetuning Qwen models on DeepMath-103K (He et al., 2025) dataset via Zero RL. These well-trained model has achieved state-of-the-art results on many challenging math benchmarks and demonstrates prominent "aha moment" phenomenon (e.g. longer response length and more cognitive behaviors).

**Metric**   Following Deepseek-R1 (Guo et al., 2025), we define a rule-based outcome verifier and report the pass@k score (Chen et al., 2021). We set the maximum generation length to 32,768 tokens. For each problem, we generate $n$ samples ($n \geqslant k$) using a sampling temperature of 0.6 and a top-p value of 0.95. Let $c$ be the number of correct samples among the $n$ generated samples, then Pass@k is calculated as:

$$\text{pass@k} = 1 - \frac{\binom{n-c}{k}}{\binom{n}{k}}. \tag{1}$$

**Evaluation**   To better understand how response length impacts performance, we developed a refined evaluation strategy. We standardized the lengths of all sampled 8,192 responses for a given problem (refer to Section 4.2) and sorted them accordingly. These responses were then uniformly divided into 16 bins based on their standardized lengths. For each bin, we calculated both the average response length and the average pass@k score. Specifically, we reported pass@1 (our general test-time metric) and pass@32 (the sampling group size used in our later training settings).

---

[1] https://huggingface.co/zwhe99/DeepMath-Zero-3B
[2] https://huggingface.co/zwhe99/DeepMath-Zero-7B

## 3.2 RESULTS

Figure 1 plots the results of pass@k with respect to the standardized response length. For test-time metrics pass@1, shorter responses exhibit better performance compared to the longer ones. However, when it comes to pass@k, longer responses surprisingly catch up and surpass their shorter counterparts, except for `DeepMath-Zero-7B` on the challenging AIME25, where we found the conclusion can still hold with a larger k value (e.g., k=64). In prevalent RL algorithms like GRPO, we usually sample multiple solutions for a single question (e.g., 32) and optimize the policy by leveraging relative comparisons between solutions. Therefore, the trend of pass@k score can be a critical guidance for these RL algorithms. On one hand, it suggests that **longer responses contain a wider coverage of potentially correct solutions**, thereby providing critical positive reward signals necessary for effective RL training. On the other hand, current length reduction strategies (Team et al., 2025) that constantly optimize for shorter responses, while seemingly improving efficiency, may inadvertently constrain the problem-solving capacity of LRMs, especially for complex problems requiring extended reasoning. However, it is inpractical to encourage LRMs to always favor longer responses in RL training processes considering the efficiency for both training and inference, indicating the need for an adaptive strategy.

## 4 DEEPCOMPRESS

We propose **DeepCompress**, a novel framework that can dynamically adjust the preference of LRMs for longer or shorter responses, in order to achieve superior performance and efficiency simultaneously. Our method enhances the Zero RL by introducing two core innovations: 1) Dual Length Reward and 2) Model-Aware Difficulty.

### 4.1 ZERO RL

In our study, we follow the zero RL training recipe from He et al. (2025) and utilize DAPO (Yu et al., 2025a) as our RL algorithm. Let $\pi_\theta$ denote the large language model policy. Given a training set $D = \{(x_i, y_i)\}$ comprising question-answer pairs where $x_i$ is a question and $y_i$ is its ground-truth answer, the language model $\pi_\theta$ samples a group of outputs $\{\hat{y}_i^1, \hat{y}_i^2, ..., \hat{y}_i^G\}$ for each question $x_i$, where $\hat{y}$ is the predicted answer and $G$ is the group size. We adopt a rule-based verifier $V$ to judge each answer, and use its final accuracy as the outcome reward. This binary reward $R_o$ is computed as:

$$R_o(\hat{y}, y) = \begin{cases} +1, & \text{if the extracted final answer is exactly correct,} \\ -1, & \text{otherwise.} \end{cases} \tag{2}$$

### 4.2 DUAL LENGTH REWARD

Our primary objective is to train models to generate correct solutions using the minimal number of tokens, thereby maximizing response efficiency. To simultaneously maintain the models' capability for deep exploration when addressing complex problems, we design distinct length reward modes for "Simple" and "Hard" questions, respectively.

Specifically, for a set of $G$ generated responses $\{\hat{y}_i^j\}_{j=1}^G$ corresponding to a given question, we compute the **response length** mean $\mu_i$ and standard deviation $\sigma_i$. The standardized length $z_i$ is then obtained as:

$$z_i = \frac{|\hat{y}_i| - \mu_i}{\sigma_i + \epsilon}, \tag{3}$$

where $\epsilon$ is a small constant introduced for numerical stability to avoid division by zero. The length reward $R_z$ utilizes a sigmoid function for nonlinear transformation:

$$R_z(\hat{y}, \beta) = \text{sigmoid}(-\beta z_i) = \frac{1}{1 + e^{\beta z_i}}, \tag{4}$$

where $\beta$ is a hyperparameter controlling the steepness of the sigmoid function, thereby modulating $R_z(\hat{y}, \beta)$'s sensitivity to token length deviations.

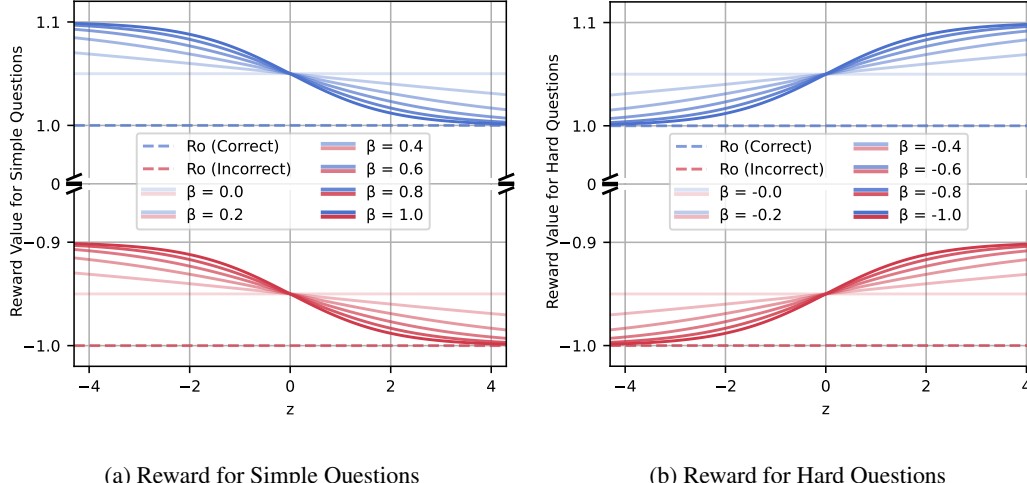

(a) Reward for Simple Questions    (b) Reward for Hard Questions

Figure 2: Reward values for our DeepCompress method. Subfigure (a) illustrates the reward for Simple Questions, and (b) for Hard Questions. For both, Blue indicates correct responses and Red indicates incorrect responses. The dashed line denotes the baseline outcome reward ($R_o$), while the solid line represents our final combined reward ($R = R_o + R_l$), effectively showcasing how our Dual Length Reward ($R_l$) dynamically modulates the reward signal based on standardized response length ($z$) and question difficulty ($\beta$).

By configuring the sign of $\beta$, we enable two operational modes as illustrated in Figure 2: 1) For simple questions, $\beta > 0$ yields higher rewards for shorter responses. 2) For complex (hard) questions, $\beta < 0$ encourages longer responses, facilitating more extensive exploration. As discussed in Section 3, this strategy increases the probability of generating at least one correct solution for difficult problems. Furthermore, we introduce a hyperparameter $\alpha$ to scale the magnitude of $R_z(\hat{y}, \beta)$, resulting in the final length reward:

$$R_l = \alpha \times R_z(\hat{y}, \beta). \tag{5}$$

## 4.3 MODEL-AWARE DIFFICULTY

A core challenge in implementing the dual length reward is the dynamic assessment of question difficulty. While public datasets like MATH (Hendrycks et al., 2021b) and DeepMath (He et al., 2025) offer curated difficulty labels, this approach incurs additional annotation costs and fails to adapt to the model's evolving capabilities during training.

To address this limitation, we propose a model-aware difficulty mechanism that dynamically classifies each question as "Simple" or "Hard". During RL training, we compute two key metrics: the **group pass ratio** per question and the **batch pass ratio**. The latter provides a real-time indicator of the model's current overall capability.

Specifically, for each question $x_i$ within a batch, we define its group pass ratio $P_g(x_i)$ as the proportion of correct responses among its $G$ generated outputs:

$$P_g(x_i) = \frac{\sum_{j=1}^{G} \mathbb{I}(R_o(\hat{y}_i^j, y_i) = 1)}{G}, \tag{6}$$

where $\mathbb{I}(\cdot)$ denotes the indicator function and $R_o$ represents the outcome reward. Concurrently, the batch pass ratio $P_b$ for a batch size $B$ is computed as:

$$P_b = \frac{\sum_{i=1}^{B} P_g(x_i)}{B}. \tag{7}$$

Here, $P_b$ quantifies the model's current global performance, while $P_g(x_i)$ reflects the difficulty of each question $x_i$ for that model state. We then determine the difficulty label by assigning $\beta$ the

following bias term, and obtain the corresponding length reward:

$$\begin{aligned} \beta_i &= P_g(x_i) - P_b \quad \in (-1, 1), \\ R_l &= \alpha \times R_z(\hat{y}, \beta_i). \end{aligned} \tag{8}$$

A positive $\beta_i$ (i.e., $P_g(x_i) > P_b$) indicates that the question is relatively easier for the current model ("Simple"). Conversely, a negative $\beta_i$ (i.e., $P_g(x_i) < P_b$) signifies that the model finds the question comparatively challenging ("Hard"). This mechanism prioritizes extended reasoning paths for the most challenging questions per batch, thereby enhancing solution coverage and ultimately improving overall performance.

Finally, the reward for RL optimization integrates both outcome reward and length reward:

$$R = R_o + R_l. \tag{9}$$

---

**Mechanism of $\beta$ in DeepCompress**

Here's a breakdown of how $\beta$ (i.e., $\beta = P_g(x_i) - P_b$) influences the length reward:

- **Mode Control:** The sign ($\pm$) of $\beta$ controls the length reward modes: 1) $\beta > 0$ designates *Simple* questions, activating short-response prioritization; 2) $\beta < 0$ flags *Hard* questions, triggering extended-reasoning mode.
- **Policy Intensity:** The absolute value ($|\beta|$) scales reward pressure, such that a larger $|\beta|$ leads a stronger preference to shorter or longer responses.

---

### 4.4 ENHANCING ROBUSTNESS

Building upon the DeepCompress framework, we introduce two enhancements to ensure robust exploration: 1) Correctness-Conditioned Lentgh Reward; 2) Smoothed Batch Pass Ratio.

**Correctness-Conditioned Length Reward**  In DeepCompress, the dual length reward applies uniformly to all generated responses, indenpendent of their correctness. This unconditional application risks creating a reward hacking scenario, where models may prioritize length optimization over solution accuracy, potentially favoring incorrect responses. To address this issue, we refine the length reward mechanism by restricting its application exclusively to responses that produce correct solutions (i.e., those with $R_o = 1$). The overall reward then becomes:

$$R = \begin{cases} R_o + R_l, & \text{if solution } y_i \text{ is correct,} \\ R_o, & \text{otherwise.} \end{cases} \tag{10}$$

**Smoothed Batch Pass Ratio**  In DeepCompress, we use the batch pass ratio $P_b$ to quantify the model's current global performance. However, this choice may impact the training stability. First, the batch pass ratio reflects only one-sided performance of the model and can fluatuate noticeably accross the batches. Second, the models often exhibit weak performance at the early steps of RL training, resulting in a low batch pass ratio. Consequently, questions may be inadvertently misjudged as simple, with undesirably large $\beta$ values (derived from $P_g - P_b$). This phenomenon can prematurely constrain response length, impeding critical exploration of the solution space.

To enhance the training robustness, we smooth the batch pass ratio by tracking its historical values with an exponential moving average (EMA). Specfically, the smoothed batch pass ratio $P_{b,t}$ at each training step $t$ is updated as:

$$P_{b,t} = \lambda \cdot P_{b,t-1} + (1 - \lambda) \cdot P_{b,t}^{true}, \tag{11}$$

where $P_{b,t}^{true}$ denotes the true batch pass ratio, and $\lambda$ ($\in [0, 1]$) is the EMA parameter. Then, $P_{b,t}$ is used in Equation 8 to determine the effective $\beta$ for length modulation. This updating rule avoids the bias by $P_{b,t}^{true}$ and gives a more stable estimation of model's current global performance. Besides, we initialize $P_{b,t}$ with 1.0, which ensures that $P_{b,t}$ gradually adapts from an optimistic initial state, preventing premature over-penalization due to a low true $P_{b,t}^{true}$ in early training.

Table 1: Math reasoning performance. "DeepCompress" denotes models trained with our novel DeepCompress approach, which significantly improves the reasoning accuracy.

| Model | MATH 500 | AMC 23 | Olympiad Bench | Minerva Math | AIME 24 | AIME 25 | Poly Math | Avg Acc |
|---|---|---|---|---|---|---|---|---|
| Qwen-2.5-3B | 50.4 | 24.2 | 21.2 | 20.4 | 4.2 | 1.5 | 21.9 | 20.5 |
| Qwen-2.5-3B-Instruct | 66.0 | 42.5 | 29.4 | 28.9 | 5.4 | 2.5 | 27.3 | 28.9 |
| DeepMath-Zero-3B | 72.8 | 48.0 | 38.0 | 30.8 | 11.5 | 6.9 | 34.1 | 34.6 |
| DeepCompress-Zero-3B | **75.3** | **49.4** | **39.3** | **32.7** | **16.7** | **7.1** | **35.8** | **36.6** |
| Qwen-2.5-7B | 54.8 | 35.3 | 27.8 | 16.2 | 7.7 | 5.4 | 28.1 | 25.0 |
| Open-Reasoner-Zero-7B | 81.8 | 58.9 | 47.9 | 38.4 | 15.6 | 14.4 | 40.7 | 42.5 |
| Qwen-2.5-7B-SRL-Zoo | 77.0 | 55.8 | 41.0 | 41.2 | 15.6 | 8.7 | 33.1 | 38.9 |
| DeepMath-Zero-7B | 85.6 | 64.7 | 51.3 | 45.4 | 19.4 | 13.1 | 42.6 | 46.0 |
| DeepCompress-Zero-7B | **85.6** | **67.8** | **53.3** | **47.4** | **23.5** | **19.6** | **44.0** | **48.7** |

## 5 EXPERIMENTS

This section details the experimental setup and presents a comprehensive evaluation across a suite of challenging mathematical benchmarks. In particular, we aim to investigate how DeepCompress improves both the performance and efficiency of models simultaneously.

### 5.1 EXPERIMENTAL SETUP

**RL Training** Our models are fine-tuned following the RL training recipe from He et al. (2025), which has produced state-of-the-art reasoning models (e.g., `DeepMath-Zero-7B`). We applied dynamic sampling policy optimization (DAPO) algorithm from Yu et al. (2025a), and trained `Qwen2.5-3B`[3], `Qwen2.5-7B`[4] with a rule-based reward $R_o$ (as defined in Equation 2). Following Hu et al. (2025), we adjusted the chat template of the Qwen model. Further details on the training settings can be found in Appendix B.

**Evaluation** We comprehensively evaluate model performance on seven challenging mathematical benchmarks: MATH-500 (Hendrycks et al., 2021b), AMC 2023 (MAA, b), OlympiadBench (He et al., 2024), Minerva Math (Lewkowycz et al., 2022), AIME 2024-2025 (MAA, a), and the English subset of PolyMath (Wang et al., 2025a). As primary metrics, we samples 16 responses for each question and report the pass@1 accuracy. We construct a validation set, which consists of 60 questions from MATH and 60 from AIME 2022-2023, to select the checkpoint with the highest pass@1 score for evaluation. We utilized vLLM (Kwon et al., 2023) for efficient batch inference, and fixed the decoding parameters to temperature=0.6, top_p=0.95, and max_tokens=32,768. To ensure fair comparison and eliminate variance from evaluation scripts, we re-evaluate all baseline models under our precise evaluation settings.

### 5.2 MAIN RESULTS

**DeepCompress exhibits stronger reasoning capabilities** Table 1 presents the main experimental results. Our proposed DeepCompress consistently outperforms all existing Zero RL baselines across all seven mathematical reasoning benchmarks, establishing a new **state-of-the-art** (SOTA). Compared to the previous SOTA model, DeepMath-Zero, DeepCompress achieves an average absolute improvement of +2.0 points with the 3B model and +2.7 points with the 7B model. Notably, DeepCompress demonstrates substantial gains on challenging problems. For example, DeepCompress-Zero-7B surpasses DeepMath-Zero-7B by +4.1 absolute points on AIME 24 and by +6.5 on AIME 25. These results highlight the robustness of DeepCompress in enhancing LLM reasoning through deeper exploration, particularly on complex tasks that require extended reasoning paths to push the boundaries of performance.

---

[3] https://huggingface.co/Qwen/Qwen2.5-3B
[4] https://huggingface.co/Qwen/Qwen2.5-7B

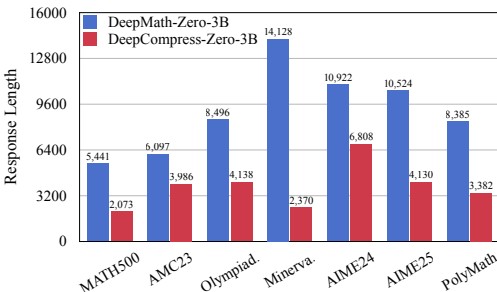 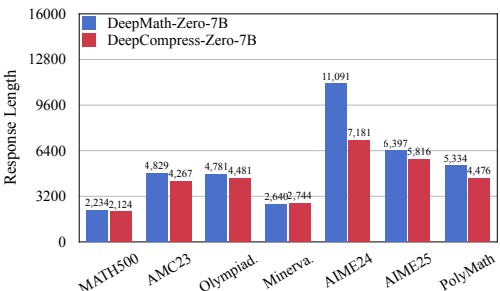

Figure 3: Average Response Length across mathematical benchmarks. DeepCompress-Zero models achieve significantly shorter average outputs compared to DeepMath-Zero models.

Table 2: Performance on the GPQA-Diamond, MMLU-STEM and Big-Bench Hard.

| Model | GPQA-Diamond | | | | MMLU | Big-Bench |
| | Biology | Physics | Chemistry | Overall | STEM | Hard |
|---|---|---|---|---|---|---|
| Qwen-2.5-3B | 29.9 | 19.8 | 20.3 | 21.0 | 7.5 | 48.1 |
| Qwen-2.5-3B-Instruct | 45.1 | 26.1 | 30.7 | 29.9 | 48.7 | 71.3 |
| DeepMath-Zero-3B | 45.1 | 25.3 | 32.2 | 30.2 | 54.7 | 71.6 |
| DeepCompress-Zero-3B | 44.1 | 25.5 | 35.7 | 31.7 | 56.0 | 73.7 |
| Qwen-2.5-7B | 33.6 | 21.4 | 27.8 | 25.3 | 12.1 | 41.5 |
| Open-Reasoner-Zero-7B | 50.3 | 26.7 | 47.8 | 38.1 | 47.0 | 83.2 |
| Qwen-2.5-7B-SimpleRL-Zoo | 31.9 | 22.6 | 37.9 | 30.2 | 15.0 | 74.9 |
| DeepMath-Zero-7B | 58.6 | 29.5 | 53.2 | 42.6 | 72.7 | 85.0 |
| DeepCompress-Zero-7B | 57.6 | 31.2 | 58.2 | 43.9 | 75.5 | 85.7 |

**DeepCompress demonstrates higher inference efficiency** As shown in Figure 3, DeepCompress generates significantly more concise responses compared to DeepMath-Zero models across all the evaluated benchmarks. On average, DeepCompress compresses the response length by 57.9% with the 3B model and 16.6% with the 7B model. Particularly on AIME 24, DeepCompress-Zero-3B uses 37.6% less tokens to achieves +5.2 absolute improvement, and DeepCompress-Zero-7B uses 35.2% less tokens to gain +4.1 improvement (see Table 1). These results establish that dynamically adjusting the preference for shorter or longer responses is truly an effective strategy for advancing reasoning boundaries while minimizing inference costs.

**Generalizable reasoning beyond mathematics** To validate that our method generalizes beyond mathematical reasoning, we added evaluations on GPQA-Diamond (Rein et al., 2024) (biology, physics and chemistry), Big Bench Hard (Hendrycks et al., 2021a) (complex multi-step reasoning), and MMLU-STEM (Suzgun et al., 2023) (science and engineering reasoning). As shown in Table 2, DeepCompress models consistently outperform other baselines, demonstrating significant improvements across all three benchmarks. These findings suggest that the dynamic strategy of exploration and efficiency facilitated by DeepCompress does not merely optimize for mathematical tasks, but rather fosters a more robust reasoning capability that generalizes across diverse scientific domains.

## 5.3 IMPACT OF LENGTH REWARD

**Experimental Settings** To understand how DeepCompress's length reward contributes to performance gains, we conducted further ablation experiments beyond the main results. Specifically, we analyzed variants trained with a fixed parameter $\beta$: **Length Penalty** ($\beta = 1$) designed purely to reduce response length, and **Length Bonus** ($\beta = -1$) intended to encourage longer outputs. Their behaviors were then compared against our DeepCompress method, alongside DeepMath-Zero-7B. We observed and analyzed policy entropy, response length, and pass@1 scores across different steps, as presented in Figure 4.

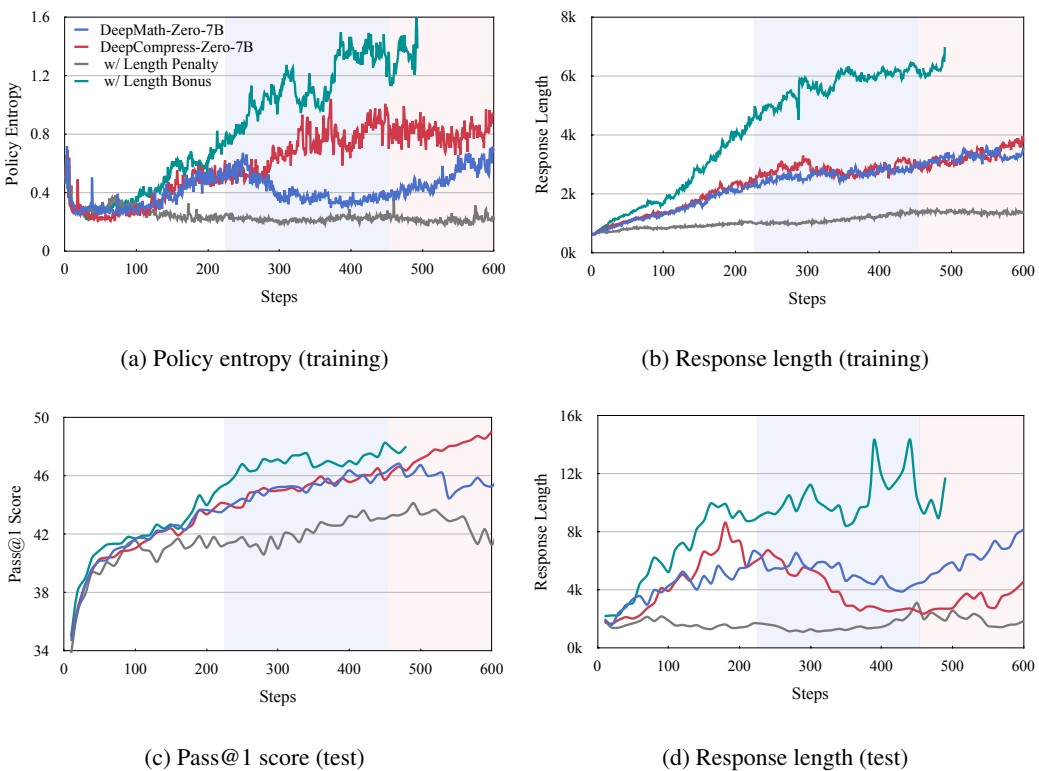

Figure 4: Training dynamics and evaluation results of DeepCompress. (a) Policy entropy during training. (b) Average response length on training batches. (c) Average pass@1 score (%) on test sets. (d) Average response length on test sets.

**Results and Analysis** As shown in Figure 4a, the policy entropy dynamics reveal interesting patterns. Models trained with length bonus exhibit higher policy entropy during training, while length penalty consistently maintains a remarkably stable and low entropy. Regarding response length as shown in Figure 4d, the models' behaviors align perfectly with our length reward design: those trained with length bonus consistently generate longer outputs, while length penalty variants produce remarkably shorter responses. Meanwhile, these additional reasoning overheads contribute to stronger mathematical reasoning capabilities. Figure 4c shows that length bonus variant exhibits higher performance compared to other methods.

In contrast, our DeepCompress method demonstrates a more adaptive balancing act. As depicted, DeepCompress's policy entropy initially increases, reflecting a phase of broad exploration, then gradually stabilizes as the model converges. Similarly, its average response length shows an initial increase (for exploration) followed by a controlled reduction (for efficiency). Throughout this dynamic process, DeepCompress's performance on test sets exhibits continuous growth. This illustrates how DeepCompress intelligently and automatically balances exploration with efficiency, achieving simultaneous optimality in both dimensions and continuously improving performance.

## 5.4 QUANTIFYING EMERGENCE OF REASONING BEHAVIORS

To further investigate the mechanisms behind DeepCompress's high policy entropy and superior performance, we conducted an analysis on a targeted set of challenging problems. Specifically, we constructed this hard problem set from instances where our baseline models (Qwen2.5-3B and Qwen2.5-7B), failed to produce a correct solution. On each set, we follow Zeng et al. (2025) to track the emergence of four cognitive behaviors described in Gandhi et al. (2025), with the prompt shown in Appendix D. The manifestation of these behaviors suggests a reproduction of the "aha moment" phenomenon observed in R1 (Guo et al., 2025).

As shown in Table 3, DeepCompress reflects more often than the baseline models. Intriguingly, despite this higher reflection frequency, its average response length remains shorter. This indicates that DeepCompress has learned a more efficient reflection process, enabling more concise and targeted attempts at a solution. This mechanism also proves highly effective, as evidenced by DeepCompress's stronger pass@1 score. Therefore, DeepCompress does not just encourage more thinking, but rather smarter thinking, turning each reflective act into a productive step towards the solution.

Table 3: Reflection Frequency on hard questions. We use GPT-4o to extract and track "aha moment" behaviors, with the prompt shown in Appendix D.

| Model | Reflect | Length | Pass@1 |
|---|---|---|---|
| DeepMath-Zero-3B | 2.45 | 11,222 | 7.21 |
| DeepCompress-Zero-3B | 2.73 | 4,853 | 8.72 |
| DeepMath-Zero-7B | 2.59 | 7,180 | 11.35 |
| DeepCompress-Zero-7B | 2.64 | 5,942 | 13.81 |
| w/ Length Penalty | 2.20 | 2,520 | 9.94 |
| w/ Length Bonus | 2.87 | 13,575 | 11.89 |

### 5.5 HYPERPARAMETER ANALYSIS

**Reward Weight $\alpha$ and EMA Parameter $\lambda$** As illustrated in Figure 5, our method exhibits remarkable robustness with respect to the reward weight $\alpha$, with all configurations consistently outperforming the previous baseline, `DeepMath-Zero-3B`. Additionally, empirical results indicate that a larger smoothing factor ($\lambda = 0.99$) is crucial for achieving optimal performance. By incentivizing longer reasoning trajectories and deeper exploration during the early training phase, this elevated factor effectively empowers the model to break through existing performance bottlenecks.

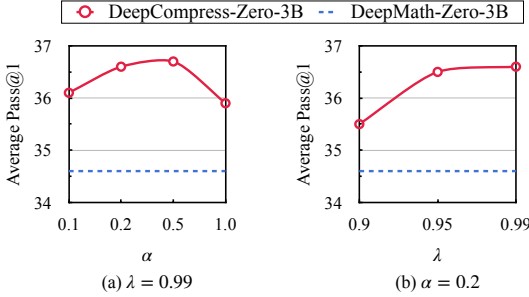

Figure 5: Ablation studies on $\alpha$ and $\lambda$. The dashed line represents the DeepMath-Zero-3B baseline.

## 6 CONCLUSION

This paper introduces **DeepCompress**, a novel framework that enhances both the accuracy and efficiency of Large Reasoning Models. By incorporating a model-aware difficulty mechanism and a dual length reward, DeepCompress dynamically adapts its reasoning strategy - encouraging concise solutions for simple problems while promoting deeper exploration for hard ones. Our experiments on challenging mathematical benchmarks show that DeepCompress achieves new state-of-the-art performance while simultaneously making significant gains in token efficiency. Further analysis reveals that our method fosters a more effective learning process by encouraging high policy entropy for exploration, leading to more frequent yet more effective reflection behaviors. By enabling models to intelligently allocate their reasoning efforts, DeepCompress represents a promising step toward developing more powerful and efficient autonomous reasoners.

A limitation of this work is that our method's effectiveness relies on sufficient length variation among responses sampled within the RL group. Furthermore, to ensure training efficiency, we capped the maximum generation length at 10k tokens. This constraint may have restricted the model's ability to explore more complex or longer-form solutions.

## 7 ETHICS STATEMENT

The authors of this work have read and adhere to the ICLR Code of Ethics. Our research focuses on developing a novel reinforcement learning framework, DeepCompress, aimed at enhancing the reasoning capabilities and computational efficiency of Large Reasoning Models. The primary goal of our work is to advance the scientific understanding of AI reasoning and to develop models that are both more effective and more efficient, which we believe is a positive step towards sustainable and accessible AI research. Our work is foundational and does not introduce societal harms. All code and models will be released publicly to promote open research and reproducibility.

## 8  REPRODUCIBILITY STATEMENT

To ensure full reproducibility, all our code, training scripts, and final model weights will be made publicly available. Detailed descriptions of our training setup, including all hyperparameters, software versions, and implementation specifics, are provided in Appendix B. This will allow researchers to verify our results, build upon our framework, and further explore adaptive training strategies for large reasoning models.

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

## A  THE USE OF LARGE LANGUAGE MODELS

In the preparation of this manuscript, we utilized a Large Language Model as a writing assistant. Its role was to aid in polishing the text, including improving grammar and sentence structure. The core research ideas, experimental design, and analysis presented in this paper are entirely our own.

## B  TRAINING DETAILS

We use `verl` as the training framework[5]. Configurations are listed in Table 4.

Table 4: Configurations for training DeepCompress series models.

| Config | DeepCompress-Zero-3B | DeepCompress-Zero-7B |
|---|---|---|
| lr | 1e-6 | 1e-6 |
| kl_coef | 0.0 | 0.0 |
| max_prompt_length | 2K | 2K |
| max_response_length | 10K | 10K |
| train_batch_size | 512 | 512 |
| ppo_mini_batch_size | 32 | 32 |
| clip_ratio_low | 0.20 | 0.20 |
| clip_ratio_high | 0.28 | 0.28 |
| temperature | 1.0 | 1.0 |
| rollout.n | 32 | 32 |
| overlong_buffer.len | 2K | 2K |
| total_training_steps | 600 | 600 |
| reward_weight $\alpha$ | 0.2 | 0.2 |
| EMA_parameter $\lambda$ | 0.99 | 0.99 |

## C  BATCH PASS RATIO

The core mechanism of DeepCompress relies on the batch pass ratio ($P_b$) to judge problem difficulty. We recorded the changes in $P_b$ throughout the training process, and as shown in Figure 6, the model's $P_b$ exhibits stable growth, indicating very low noise in the difficulty judgment process.

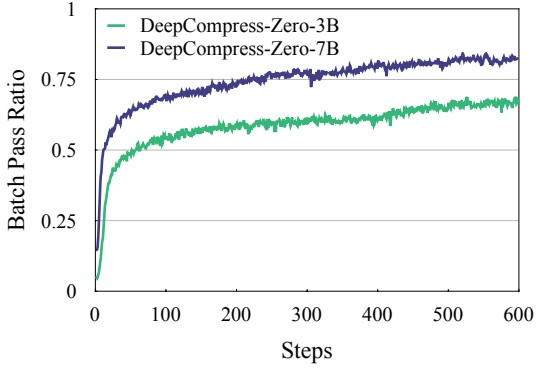

Figure 6: Batch pass ratio during training.

---

[5]https://github.com/volcengine/verl

## D    REASONING BEHAVIOR PROMPT

> **Prompt for Identifying and Analyzing Reasoning Behaviors**
>
> Below is a chain-of-reasoning generated by a Language Model when attempting to solve a math problem. Evaluate this chain-of-reasoning to determine whether it demonstrates beneficial problem-solving behaviors that deviate from typical linear, monotonic reasoning patterns commonly observed in language models.
>
> <start_of_reasoning>
> {input}
> <end_of_reasoning>
>
> Specifically, actively identify and emphasize beneficial behaviors such as:
> (1) Backtracking: Explicitly revising approaches upon identifying errors or dead ends
> (e.g., "This approach won't work because...").
>
> (2) Verification: Systematically checking intermediate results or reasoning steps
> (e.g., "Let's verify this result by...").
>
> (3) Subgoal Setting: Breaking down complex problems into smaller, manageable steps
> (e.g., "To solve this, we first need to...").
>
> (4) Enumeration: Solving problems by exhaustively considering multiple cases or possibilities.
>
> Additionally, remain attentive to and encourage the identification of other beneficial behaviors not explicitly listed here, such as creative analogies, abstraction to simpler cases, or insightful generalizations.
>
> Important:
> Clearly specify each beneficial behavior you identify.
> Provide explicit examples from the reasoning chain.
> If no beneficial behaviors are observed, explicitly return an empty list.
> Provide your evaluation clearly, formatted as follows:
>
> ```json
> {
> "behaviour": "",
> "example": ""
> }
> ```

