# OpenReview forum: "DeepCompress: A Dual Reward Strategy for Dynamically Exploring and Compressing Reasoning Chains"
_ICLR.cc/2026/Conference — ICLR 2026 Poster_

### Official Review · Reviewer_t6ev · 2025-10-30

**Soundness:** 2
**Presentation:** 2
**Contribution:** 2
**Rating:** 4
**Confidence:** 4

**Summary:**

This paper introduces DeepCompress, a novel framework that enhances Large Reasoning Models (LRMs) by dynamically classifying problems as "Simple" or "Hard" based on the model's real-time performance. By applying a dual-reward strategy that encourages shorter reasoning for "Simple" problems and longer exploration for "Hard" ones, DeepCompress achieves improved accuracy while simultaneously improving token efficiency on multiple mathematical benchmarks.

**Strengths:**

1. The problem studied in this paper is critical.
2. The paper is easy to follow and understand.
3. Experimental results show that the performance of the proposed method improves consistently over base models.

**Weaknesses:**

1. The paper lacks important baselines to compare with other RL-based reasoning chain compression works, for example [1, 2, 3], and does not position the proposed method among them.
2. The difficulty measure completely relies on batches, which may lead to instability. In a batch containing only simple problems, a problem that is simply "not that simple" may also be incorrectly marked as "difficult" because $P_g $is lower than the extremely high $P_b $. Vice versa. This mechanism cannot enable the model to learn a stable and absolute internal representation of the difficulty of the problem, but instead causes the reward signal to fluctuate dramatically with the random fluctuations of batch data.
3. It is unclear how the hyperparameters are determined, e.g., $\beta$ is set to 1 and -1 without explanation or investigation.
4. How will the performance change with a different group size $G$?

References:

[1] Tu S, Lin J, Zhang Q, et al. Learning When to Think: Shaping Adaptive Reasoning in R1-Style Models via Multi-Stage RL[J]. arXiv preprint arXiv:2505.10832, 2025.

[2] Shrivastava V, Awadallah A, Balachandran V, et al. Sample more to think less: Group filtered policy optimization for concise reasoning[J]. arXiv preprint arXiv:2508.09726, 2025.

[3] Yuan D, Xie T, Huang S, et al. Efficient RL Training for Reasoning Models via Length-Aware Optimization[J]. arXiv preprint arXiv:2505.12284, 2025.

**Questions:**

See weaknesses.

---

> ### Author Response · Authors · 2025-11-24
> **Response to Reviewer t6ev**
>
> We appreciate your thorough review and valuable feedback. We address the comments below:
>
> > W1. More length control baselines
>
> In addition to the existing Open-Reasoner-Zero and SimpleRL-Zoo baselines, we have compared against other length-penalty strategies such as those in Kimi-k1.5 [1] and Short-RL [2], with the results presented below.
>
> This finding is consistent with our preliminary experimental conclusions: while these methods achieve excellent results in length compression, their performance is simultaneously affected. DeepCompress, through its dynamic training strategy, aims to achieve the optimum performance in both efficiency and performance.
>
> | Pass@1               | MATH500 | AMC23 | Olympiad Bench | Minerva Math | AIME24 | AIME25 | Poly Math | Avg Acc |
> | -------------------- | ------- | ----- | -------------- | ------------ | ------ | ------ | --------- | ------- |
> | Kimi-k1.5            | 53.4    | 36.6  | 19.7           | 20.3         | 6.7    | 1.2    | 28.2      | 23.7    |
> | Short-RL             | 63.4    | 35.2  | 26.1           | 31.1         | 8.1    | 1.9    | 29.4      | 27.9    |
> | DeepCompress-Zero-3B | 75.3    | 49.4  | 39.3           | 32.7         | 16.7   | 7.1    | 35.8      | 36.6    |
>
> | Length               | MATH500 | AMC23 | Olympiad Bench | Minerva Math | AIME24 | AIME25 | Poly Math | Avg Len |
> | -------------------- | ------- | ----- | -------------- | ------------ | ------ | ------ | --------- | ------- |
> | Kimi-k1.5            | 162     | 102   | 274            | 182          | 70     | 222    | 80        | 156     |
> | Short-RL             | 411     | 513   | 752            | 251          | 843    | 445    | 342       | 508     |
> | DeepCompress-Zero-3B | 2073    | 3986  | 4138           | 2370         | 6808   | 4130   | 3382      | 3841    |
>
> [1] Kimi k1.5: Scaling Reinforcement Learning with LLMs
>
> [2] Efficient RL Training for Reasoning Models via Length-Aware Optimization
>
> > W2. Concern about batch pass ratio
>
> DAPO proposes to over-sample and filter out prompts with the accuracy equal to 1 and 0. In fact, our sampling batch size are four times larger than training, reaching 2048. As shown in the table below, the batch pass ratio increases very steadily during the training process.
>
> | **Batch Pass Ratio** | 20   | 40   | 60   | 80   | 100  | 120  | 140  | 160  | 180  | 200  | 220  | 240  | 260  | 280  | 300  | 320  | 340  | 360  | 380  | 400  | 420  | 440  | 460  | 480  | 500  |
> | -------------------- | ---- | ---- | ---- | ---- | ---- | ---- | ---- | ---- | ---- | ---- | ---- | ---- | ---- | ---- | ---- | ---- | ---- | ---- | ---- | ---- | ---- | ---- | ---- | ---- | ---- |
> | DeepCompress-Zero-3B | 38.4 | 47.1 | 52.2 | 53.2 | 53.9 | 56.7 | 55.3 | 57.6 | 59.4 | 58.7 | 58.3 | 59.4 | 60.5 | 59.9 | 61.3 | 61.1 | 60.8 | 62.0 | 61.2 | 60.4 | 62.2 | 63.0 | 65.5 | 66.3 | 66.8 |
> | DeepCompress-Zero-7B | 55.8 | 61.7 | 64.9 | 66.6 | 68.6 | 69.7 | 70.0 | 70.9 | 71.7 | 73.5 | 75.8 | 76.2 | 77.0 | 76.6 | 77.8 | 77.1 | 77.4 | 77.6 | 79.3 | 79.0 | 79.3 | 79.8 | 80.6 | 80.9 | 80.9 |
>
> We recognize this is a common concern among reviewers. We have added a new figure (Figure 5 in Appendix C) plotting the complete training steps to address this.
>
> > W3. How the hyperparameters $\beta$ are determined?
>
> For the DeepCompress framework, $\beta$ is **not a fixed hyperparameter**, but a value dynamically calculated by the model-aware difficulty mechanism: $\beta_{i}=P_{g}(x_{i})-P_{b}$. Its theoretical value range is $[-1, 1]$.
>
> The values $\beta=1$ (Length Penalty) and $\beta=-1$ (Length Bonus) were only used in the ablation study (Section 5.3). They were specifically chosen to analyze the **extreme boundary cases** of our dynamic dual reward strategy.
>
> > W4. How will the performance change with a different group size?
>
> Scaling group size stabilizes the learning signal through broader exploration [1]. As confirmed by our preliminary experimental results (Figure 1 and Line 167-168), DeepCompress will perform better with a larger group size. As $k$ increases, the pass@k for longer responses increases more significantly, particularly on challenging problems. DeepCompress leverages this broader coverage, allowing it to find and reinforce positive reward signals, thereby enhancing its advantage.
>
> [1] BroRL: Scaling Reinforcement Learning via Broadened Exploration

---

> ### Author Response · Authors · 2025-11-27
> **Thanks for reviewing & Additional feedback welcome**
>
> Dear reviewer,
>
> We appreciate your thorough review and the valuable feedback. We hope our responses adequately address your concerns and look forward to any additional comments you may have. We would be grateful if you can acknowledge our responses and consider updating your scores.
>
> Best,
>
> Authors

---

> > ### Comment · Reviewer_t6ev · 2025-11-27
> >
> > I appreciate the feedback that addresses most of my concerns and raises the score to 6.

---

> > > ### Author Response · Authors · 2025-11-27
> > >
> > > Thank you for your positive evaluation. We greatly appreciate your constructive engagement throughout the discussion, and we will incorporate the outcomes of this discussion into the final version.

---

### Official Review · Reviewer_GJ2N · 2025-11-01

**Soundness:** 3
**Presentation:** 2
**Contribution:** 3
**Rating:** 4
**Confidence:** 4

**Summary:**

This paper proposes DeepCompress, a reinforcement learning (RL) framework for large reasoning models (LRMs) that introduces a dual reward strategy—encouraging shorter reasoning for “simple” problems and longer reasoning for “hard” ones. The method dynamically classifies questions based on the group pass ratio within RL batches, and adjusts a sigmoid-shaped length reward accordingly. The authors claim improved accuracy and token efficiency across mathematical reasoning benchmarks such as MATH-500, AIME, and OlympiadBench.

**Strengths:**

- The problem setup—balancing reasoning efficiency and accuracy—is interesting and timely.

- The adaptive length reward (shorter for easy, longer for hard) is conceptually reasonable and aligns with observed overthinking/underthinking phenomena in LLMs.

- The paper provides extensive experimental evaluation on multiple math reasoning benchmarks.

- Figures and tables (e.g., Table 1, Fig. 4) show measurable improvements over baseline RL models like DeepMath-Zero.

**Weaknesses:**

1. Missing ablation and causal analysis: Although the authors attribute improvements to the dual reward, there are no clear ablations isolating which component (dual reward vs. model-aware difficulty) contributes most to the gain. The provided variants (length bonus / penalty) are simplistic and insufficient to explain why the accuracy improves. There’s also no sensitivity study on α, β, or λ—key hyperparameters controlling reward magnitude.

2. It seems the results contradict the motivation on length vs. accuracy: Figure 1 and Table 4(c) still show that pass@1 accuracy decreases with increasing reasoning length. This raises doubts about the claim that longer reasoning improves correctness for hard problems. The authors interpret longer reasoning as beneficial in the RL phase, but the empirical results suggest diminishing returns for long outputs.

3. Ambiguity in difficulty classification: The definition of “simple” vs. “hard” via group pass ratio is circular—model competence determines the classification, which in turn affects training rewards. This may cause instability or reinforce existing biases rather than lead to genuine adaptivity.

4. Unclear explanation of policy gradients and learning signal: It is not clear how the “dual reward” integrates into the policy gradient objective beyond heuristic addition. The paper repeatedly mentions DAPO and GRPO but doesn’t formally show how the modified reward affects gradient updates or stability. It reads more as an empirical tweak than a well-motivated RL formulation.

**Questions:**

1. How does the model perform if we train only with the model-aware difficulty component but no explicit length reward?
2. Does the adaptive β term (Eq. 8) introduce additional variance into the policy gradient estimator? If so, how do you ensure stable convergence?
3. How much of the gain comes from model-aware difficulty versus dual reward?

---

> ### Author Response · Authors · 2025-11-24
> **Response to Reviewer GJ2N (1 of 2)**
>
> We thank Reviewer GJ2N for the careful review, pointing out more analysis and ablation experiments in our work. We address the comments below:
>
> > W1. More hyper-parameter analysis
>
> Thank you for your suggestions. We add more ablation experiment accordong to different hyper-parameters $\alpha$, $\beta$ and $\lambda$.
>
> 1. **$\alpha$**: We trained the Qwen2.5-3B model under the setting where the EMA parameter $\lambda$ was fixed at $0.99$. The results below illustrate the performance achieved using different $\alpha$. **All results outperforms previous SOTA DeepMath-Zero-3B (34.6).**
>
> | alpha       | 0.1  | 0.2  | 0.5  | 1.0  |
> | ----------- | ---- | ---- | ---- | ---- |
> | **Avg Acc** | 36.1 | 36.6 | 36.7 | 35.9 |
>
> 2. **$\beta$**: We must clarify that $\beta$ is **not** a hyperparameter. Instead, it is a value dynamically calculated by the model-aware difficulty mechanism ($P_g - P_b$ in Eq.8). We have supplemented our experiments with training based on data difficulty annotations (i.e., using DeepMath's difficulty labels), and the results are shown below.
>
> | Pass@1                        | MATH500  | AMC23    | Olympiad Bench | Minerva Math | AIME24   | AIME25  | Poly Math | Avg Acc  |
> | ----------------------------- | -------- | -------- | -------------- | ------------ | -------- | ------- | --------- | -------- |
> | Data-Labeled Difficulty       | 72.7     | 43.9     | 36.0           | 31.8         | 9.4      | 5.0     | 32.7      | 33.1     |
> | Model-Aware Difficulty (Ours) | **75.3** | **49.4** | **39.3**       | **32.7**     | **16.7** | **7.1** | **35.8**  | **36.6** |
>
> 3. **$\lambda$**: We tested different $\lambda$ while fixing $\alpha$ at 0.2. **We found that selecting a slightly larger $\lambda$ is very necessary to encourage the model's early exploration.**
>
> | lambda                   | 0.9  | 0.95 | 0.99 |
> | ------------------------ | ---- | ---- | ---- |
> | **DeepCompress-Zero-3B** | 35.5 | 36.5 | 36.6 |
> | **DeepCompress-Zero-7B** | 44.5 | 48.9 | 48.7 |
>
> > W2. Misunderstanding on the relationship between length and difficulty
>
> We apologize for the imprecision. We did not discuss the relationship between difficulty and length; we clarify this as follows:
>
> 1. For any given question (**regardless of difficulty**), we sample multiple times and obtain responses of different lengths.
> 2. We found that the pass@32 metric for the set of longer responses is higher than that of the shorter set.
> 3. This inspired us to encourage the model to increase length, making it easier to obtain at least one correct answer. This positive training signal allows the RL training to proceed.
> 4. This characteristic may be more important for the model in solving hard problems.
>
> In the experimental result, Our length-bonus strategy consistently exhibited higher performance compared to other methods among the completed training steps.
>
> > W3. Concern about batch pass ratio
>
> DAPO proposes to over-sample and filter out prompts with the accuracy equal to 1 and 0. In fact, our sampling batch size are four times larger than training, reaching 2048. As shown in the table below, the batch pass ratio increases very steadily during the training process.
>
> | **Batch Pass Ratio** | 20   | 40   | 60   | 80   | 100  | 120  | 140  | 160  | 180  | 200  | 220  | 240  | 260  | 280  | 300  | 320  | 340  | 360  | 380  | 400  | 420  | 440  | 460  | 480  | 500  |
> | -------------------- | ---- | ---- | ---- | ---- | ---- | ---- | ---- | ---- | ---- | ---- | ---- | ---- | ---- | ---- | ---- | ---- | ---- | ---- | ---- | ---- | ---- | ---- | ---- | ---- | ---- |
> | DeepCompress-Zero-3B | 38.4 | 47.1 | 52.2 | 53.2 | 53.9 | 56.7 | 55.3 | 57.6 | 59.4 | 58.7 | 58.3 | 59.4 | 60.5 | 59.9 | 61.3 | 61.1 | 60.8 | 62.0 | 61.2 | 60.4 | 62.2 | 63.0 | 65.5 | 66.3 | 66.8 |
> | DeepCompress-Zero-7B | 55.8 | 61.7 | 64.9 | 66.6 | 68.6 | 69.7 | 70.0 | 70.9 | 71.7 | 73.5 | 75.8 | 76.2 | 77.0 | 76.6 | 77.8 | 77.1 | 77.4 | 77.6 | 79.3 | 79.0 | 79.3 | 79.8 | 80.6 | 80.9 | 80.9 |
>
> We recognize this is a common concern among reviewers. We have added a new figure (Figure 5 in Appendix C) plotting the complete training steps to address this.

---

> ### Author Response · Authors · 2025-11-24
> **Response to Reviewer GJ2N (2 of 2)**
>
> > W4. Explanation of policy gradients and learning signal
>
> The policy gradient is derived from the **Expected Advantage** of the policy $\pi_{\theta}$, where the total reward $R$ is defined as the sum of the Outcome Reward ($R_o$) and the Dual Length Reward ($R_l$):
>
> $$R = R_o + R_l \text{, where } R_l = \alpha \times \text{sigmoid}(\mp \beta_i z_i)$$
>
> The gradient $\nabla_\theta J(\theta)$ is calculated by applying the GRPO/DAPO policy update rule to the full, non-heuristic reward $R$. The **Learning Signal** for the optimization is the **Group Relative Advantage** ($\hat{A}_{i}$) derived from this total reward $R$:
>
> $$\hat{A}_i = \frac{R_i - \text{mean}(\{R_1, \dots, R_G\})}{\text{std}(\{R_1, \dots, R_G\})}$$
>
> The policy gradient $\nabla_{\theta} J(\theta)$ is given by:
>
> $$\nabla_{\theta} J(\theta) \approx E_{\tau \sim \pi_{\theta_{old}}} \left[ \sum_{t=1}^{T} \nabla_{\theta} \log \pi_{\theta}(\tau_t) \cdot \hat{A}_{i,t} \cdot \mathcal{I} \right]$$
>
> The dual reward is not a heuristic addition to the gradient, but a principled **reward shaping** signal. It modifies the total reward *before* it is fed into the non-linear GRPO advantage estimator.
>
> > Q1. Baseline using MATH's difficulty labels
>
> Please refer to W1. hyper-parameter analysis - $\beta$, the **Model-Aware Difficulty** (Ours) consistently outperforms the **Data-Labeled Difficulty** across all tested benchmarks.
>
> > Q2. How to ensure stable convergence
>
> In DAPO algorithm, the advantage calculation standardizes the final summed reward. In our training process, the model demonstrated stable convergence and achieved significant performance improvements.
>
> > Q3. How much of the gain comes from model-aware difficulty versus dual reward?
>
> These two components are functionally inseparable, as the Model-Aware Difficulty mechanism is the necessary input for calculating the Dual Length Reward.
>
> - The Model-Aware Difficulty mechanism dynamically classifies each question as "Simple" or "Hard" in real-time by calculating the difficulty bias ($\beta_i$).
> - The Dual Length Reward then uses this $\beta_i$ to configure its operational mode: if $\beta_i$ is positive ("Simple"), it rewards shorter responses; if $\beta_i$ is negative ("Hard"), it encourages longer responses.
>
> Experimental results on challenging mathematical benchmarks demonstrate the capability of DeepCompress in achieving superior performance consistently over baseline methods while also improving the token efficiency significantly.

---

> ### Author Response · Authors · 2025-11-27
> **Thanks for reviewing & Additional feedback welcome**
>
> Dear reviewer,
>
> We appreciate your thorough review and the valuable feedback. We hope our responses adequately address your concerns and look forward to any additional comments you may have. We would be grateful if you can acknowledge our responses and consider updating your scores.
>
> Best,
>
> Authors

---

> > ### Comment · Reviewer_GJ2N · 2025-11-27
> >
> > Thank you for the responses. Most of my concerns are addressed. I have updated the rating to 6.

---

### Official Review · Reviewer_EuSL · 2025-11-02

**Soundness:** 3
**Presentation:** 3
**Contribution:** 3
**Rating:** 6
**Confidence:** 4

**Summary:**

This paper presents an an adaptive length reward mechanism that trade-off the efficiency and performance for the large reasoning models (LRM). Different from the previous length-penality method, this method introcue the length reward according to the “difficulty” of problem at hand. However, while the core idea has merit, there are several concerns regarding experimental rigor, theoretical justification, and presentation clarity that need to be addressed.

**Strengths:**

+ Observation: The observation that Pass@1 decreases while Pass@32 increases with length (Figure 1) is genuinely insightful.

+ The experiments is fully evaluated acorss different becnhmark (7 benchmark) and is hoslitc enough for the math problem.

+ The paper presents a coherent and compelling research: clear motivation, principled method design, and  in-depth analysis. The writing clearly connects empirical findings to design choices to performance outcomes, making the contribution well-motivated.

**Weaknesses:**

1. The authors observe that Pass@1 diminishes with length increase, while Pass@32 generally increases. The paper attributes this to "longer responses contain a wider coverage of potentially correct solutions." However, this interpretation requires deeper scrutiny:
- **Is this truly about difficulty?** Or is it simply a statistical artifact? When you have 32 samples, longer responses might just have more opportunities to "stumble upon" correct solutions through increased exploration, not necessarily because the problem is inherently harder.
- What is the correlation between problem difficulty (ground truth labels from MATH dataset) and average response length? This would validate whether length truly reflects difficulty or model capability.

2. The classification depends on batch composition. A problem could be "Simple" in one batch and "Hard" in another depending on what other problems are sampled. This inconsistency could lead to unstable training signals. And moreover, there is a lot of recent study on dynamic batching and rollout filtering, the usage of batch and group-level metric for defining “difficulty” may be somewhat unrealiable.

3. Experimetns, since the author demonstrate the adaptive length rewards, I think at least should compare with some popular length-penality such as those in Kimi-1.5 or over-long filtering method. But in current experiments, the author just experimenet with several baseline models.

4. Moreover, I would recommand using Qwen-3 series for further validation since Qwen-3 is more prone to think longer comapred to Qwen-2.5.

5. I will be glad to see the ablation experiment accordong to different hyper-parameters such as $\beta$ in Eq.4, the reward weight $\alpha$.

6. In Figure 4, Length Bonus baseline shows higher entropy and longer responses but lower final performance than DeepCompress. This contradicts the paper's claim that longer responses help with hard problems. Why doesn't pure length bonus work better?

7. Typo

    + Lien 58: infeasbile -> infeasible
    + Line 59: effciency -> efficiency

**Questions:**

+ How does DeepCompress perform on problems of known, fixed difficulty (e.g., using MATH's difficulty labels)? This would separate the "model-aware" aspect from actual difficulty.

    + Or use similar difficulty assessment similar to [1]

        [1] Part I: Tricks or Traps? A Deep Dive into RL for LLM Reasoning. Zihe Liu, et al.

+ Have you tested on non-mathematical reasoning tasks? (e.g., LiveCodeBench)

---

> ### Author Response · Authors · 2025-11-24
> **Response to Reviewer EuSL (1 of 2)**
>
> We appreciate reviewer EuSL for recognizing our work and providing valuable suggestions.
>
> > W1. Misunderstanding on the relationship between length and difficulty
>
> We apologize for the imprecision. We did not discuss the relationship between difficulty and length; we clarify this as follows:
>
> 1. For any given question (**regardless of difficulty**), we sample multiple times and obtain responses of different lengths.
> 2. We found that the pass@32 metric for the set of longer responses is higher than that of the shorter set.
> 3. This inspired us to encourage the model to increase length, making it easier to obtain at least one correct answer. This positive training signal allows the RL training to proceed.
> 4. This characteristic may be more important for the model in solving hard problems.
>
> > W2. Concern about batch pass ratio
>
> DAPO proposes to over-sample and filter out prompts with the accuracy equal to 1 and 0. In fact, our sampling batch size are four times larger than training, reaching 2048. As shown in the table below, the batch pass ratio increases very steadily during the training process.
>
> | **Batch Pass Ratio** | 20   | 40   | 60   | 80   | 100  | 120  | 140  | 160  | 180  | 200  | 220  | 240  | 260  | 280  | 300  | 320  | 340  | 360  | 380  | 400  | 420  | 440  | 460  | 480  | 500  |
> | -------------------- | ---- | ---- | ---- | ---- | ---- | ---- | ---- | ---- | ---- | ---- | ---- | ---- | ---- | ---- | ---- | ---- | ---- | ---- | ---- | ---- | ---- | ---- | ---- | ---- | ---- |
> | DeepCompress-Zero-3B | 38.4 | 47.1 | 52.2 | 53.2 | 53.9 | 56.7 | 55.3 | 57.6 | 59.4 | 58.7 | 58.3 | 59.4 | 60.5 | 59.9 | 61.3 | 61.1 | 60.8 | 62.0 | 61.2 | 60.4 | 62.2 | 63.0 | 65.5 | 66.3 | 66.8 |
> | DeepCompress-Zero-7B | 55.8 | 61.7 | 64.9 | 66.6 | 68.6 | 69.7 | 70.0 | 70.9 | 71.7 | 73.5 | 75.8 | 76.2 | 77.0 | 76.6 | 77.8 | 77.1 | 77.4 | 77.6 | 79.3 | 79.0 | 79.3 | 79.8 | 80.6 | 80.9 | 80.9 |
>
> We recognize this is a common concern among reviewers. We have added a new figure (Figure 5 in Appendix C) plotting the complete training steps to address this.
>
> > W3. More length control baselines
>
> In addition to the existing Open-Reasoner-Zero and SimpleRL-Zoo baselines, we have compared against other length-penalty strategies such as those in Kimi-k1.5 [1] and Short-RL [2], with the results presented below.
>
> This finding is consistent with our preliminary experimental conclusions: while these methods achieve excellent results in length compression, their performance is simultaneously affected. DeepCompress, through its dynamic training strategy, aims to achieve the optimum performance in both efficiency and performance.
>
> | Pass@1               | MATH500 | AMC23 | Olympiad Bench | Minerva Math | AIME24 | AIME25 | Poly Math | Avg Acc |
> | -------------------- | ------- | ----- | -------------- | ------------ | ------ | ------ | --------- | ------- |
> | Kimi-k1.5            | 53.4    | 36.6  | 19.7           | 20.3         | 6.7    | 1.2    | 28.2      | 23.7    |
> | Short-RL             | 63.4    | 35.2  | 26.1           | 31.1         | 8.1    | 1.9    | 29.4      | 27.9    |
> | DeepCompress-Zero-3B | 75.3    | 49.4  | 39.3           | 32.7         | 16.7   | 7.1    | 35.8      | 36.6    |
>
> | Length               | MATH500 | AMC23 | Olympiad Bench | Minerva Math | AIME24 | AIME25 | Poly Math | Avg Len |
> | -------------------- | ------- | ----- | -------------- | ------------ | ------ | ------ | --------- | ------- |
> | Kimi-k1.5            | 162     | 102   | 274            | 182          | 70     | 222    | 80        | 156     |
> | Short-RL             | 411     | 513   | 752            | 251          | 843    | 445    | 342       | 508     |
> | DeepCompress-Zero-3B | 2073    | 3986  | 4138           | 2370         | 6808   | 4130   | 3382      | 3841    |
>
> [1] Kimi k1.5: Scaling Reinforcement Learning with LLMs
>
> [2] Efficient RL Training for Reasoning Models via Length-Aware Optimization

---

> ### Author Response · Authors · 2025-11-24
> **Response to Reviewer EuSL (2 of 2)**
>
> > W4. More hyper-parameter analysis
>
> Thank you for your suggestions. We add more ablation experiment accordong to different hyper-parameters $\alpha$, $\beta$ and $\lambda$.
>
> 1. **$\alpha$**: We trained the Qwen2.5-3B model under the setting where the EMA parameter $\lambda$ was fixed at $0.99$. The results below illustrate the performance achieved using different $\alpha$. **All results outperforms previous SOTA DeepMath-Zero-3B (34.6).**
>
> | alpha       | 0.1  | 0.2  | 0.5  | 1.0  |
> | ----------- | ---- | ---- | ---- | ---- |
> | **Avg Acc** | 36.1 | 36.6 | 36.7 | 35.9 |
>
> 2. **$\beta$**: We must clarify that $\beta$ is **not** a hyperparameter. Instead, it is a value dynamically calculated by the model-aware difficulty mechanism ($P_g - P_b$ in Eq.8). We have supplemented our experiments with training based on data difficulty annotations (i.e., using DeepMath's difficulty labels), and the results are shown below.
>
> | Pass@1                        | MATH500  | AMC23    | Olympiad Bench | Minerva Math | AIME24   | AIME25  | Poly Math | Avg Acc  |
> | ----------------------------- | -------- | -------- | -------------- | ------------ | -------- | ------- | --------- | -------- |
> | Data-Labeled Difficulty       | 72.7     | 43.9     | 36.0           | 31.8         | 9.4      | 5.0     | 32.7      | 33.1     |
> | Model-Aware Difficulty (Ours) | **75.3** | **49.4** | **39.3**       | **32.7**     | **16.7** | **7.1** | **35.8**  | **36.6** |
>
> 3. **$\lambda$**: We tested different $\lambda$ while fixing $\alpha$ at 0.2. **We found that selecting a slightly larger $\lambda$ is very necessary to encourage the model's early exploration.**
>
> | lambda                   | 0.9  | 0.95 | 0.99 |
> | ------------------------ | ---- | ---- | ---- |
> | **DeepCompress-Zero-3B** | 35.5 | 36.5 | 36.6 |
> | **DeepCompress-Zero-7B** | 44.5 | 48.9 | 48.7 |
>
> > W5. Why doesn't pure length bonus work better?
>
> The Length-Bonus strategy consistently exhibited higher performance compared to other methods among the completed training steps. However, after step 480, excessively long generation led to frequent Out-Of-Memory errors, forcing us to terminate this Length-Bonus experiment.
>
> On the other hand, this is precisely where the core novelty of our dynamic strategy lies. DeepCompress teaches the model through the following steps:
>
> ​	1）For hard problems, it encourages longer responses to reinforce the ability to explore correct answers, promoting generalization unsolvable problem sets.
>
> ​	2）Once a correct answer is sampled, the model receives a positive training signal that reinforces its mathematical reasoning capabilities.
>
> ​	3）As the model's capability improves and it successfully solves more problems, it learns to solve efficiently.
>
> Throughout the **Exploration-to-Exploitation** process, DeepCompress "recalls" problems that originally yielded no positive training signals.
>
> > Q1. Baseline using MATH's difficulty labels
>
> Please refer to W4. hyper-parameter analysis - $\beta$, the **Model-Aware Difficulty** (Ours) consistently outperforms the **Data-Labeled Difficulty** across all tested benchmarks.
>
> > Q2. More non-mathematical reasoning tasks
>
> Thanks for your suggestion. We conducted additional evaluations on three benchmarks: GPQA-Diamond (biology, physics and chemistry), Big Bench Hard (BBH) (complex multi-step reasoning), and MMLU-STEM (science/engineering reasoning).
>
> The results show that DeepCompress serise models consistently outperforms all baselines:
>
> | Pass@1                   | GPQA     | BBH      | **MMLU-STEM** |
> | ------------------------ | -------- | -------- | ------------- |
> | Qwen-2.5-3B              | 21.0     | 7.5      | 48.1          |
> | Qwen-2.5-3B-Instruct     | 29.9     | 48.7     | 71.3          |
> | DeepMath-Zero-3B         | 30.2     | 54.7     | 71.6          |
> | **DeepCompress-Zero-3B** | **31.7** | **56.0** | **73.3**      |
> | Qwen-2.5-7B              | 25.3     | 12.1     | 41.3          |
> | Open-Reasoner-Zero-7B    | 38.1     | 47.0     | 83.2          |
> | Qwen-2.5-7B-SimpleRL-Zoo | 30.2     | 15.0     | 74.9          |
> | DeepMath-Zero-7B         | 42.6     | 72.7     | 85.0          |
> | **DeepCompress-Zero-7B** | **43.9** | **75.5** | **85.7**      |

---

> ### Author Response · Authors · 2025-11-27
> **Thanks for reviewing & Additional feedback welcome**
>
> Dear reviewer,
>
> We appreciate your thorough review and the valuable feedback. We hope our responses adequately address your concerns and look forward to any additional comments you may have. We would be grateful if you can acknowledge our responses and consider updating your scores.
>
> Best,
>
> Authors

---

### Official Review · Reviewer_kWZ8 · 2025-11-04

**Soundness:** 2
**Presentation:** 3
**Contribution:** 2
**Rating:** 4
**Confidence:** 3

**Summary:**

This paper introduces DeepCompress, a new framework designed to enhance both the accuracy and efficiency of Large Reasoning Models. The authors identify that LRMs often "overthink" simple problems and "underthink" complex ones, and that current methods to shorten reasoning paths often sacrifice accuracy.

DeepCompress addresses this with a dual reward strategy in an RL setup. It dynamically classifies problems as "Simple" or "Hard" based on the model's real-time performance. For "Simple" problems, it rewards shorter, more efficient reasoning chains. For "Hard" problems, it encourages longer, more exploratory thought processes.

Experimental results on challenging mathematical benchmarks show that models trained with DeepCompress consistently outperform baselines, achieving higher accuracy while simultaneously generating more concise responses, thus improving token efficiency.

**Strengths:**

- This paper introduces a method that rewards shorter reasoning for simple problems and encourages longer, exploratory reasoning for difficult ones. This challenges the standard approach of always favoring conciseness.

- This paper demonstrates that DeepCompress models achieve state-of-the-art results, outperforming strong baseline models across multiple challenging mathematical reasoning benchmarks, especially on difficult problems like AIME.

- The research provides a thorough analysis showing that DeepCompress encourages a more effective learning process by balancing exploration (via high policy entropy) and efficiency, leading to more frequent and effective "reflection" behaviors.

**Weaknesses:**

- The core mechanism hinges on classifying a problem as "Simple" or "Hard" based on whether its group pass ratio (Pg) is above or below the batch pass ratio (Pb). This is a noisy and relative metric. A problem isn't inherently "Hard"; it's just harder than the batch average. This could lead to suboptimal rewards, especially in batches with skewed difficulty distributions.

- The paper proposes conditioning the length reward on the correctness of a solution to prevent reward hacking. However, this creates a paradox for the most challenging problems. If a problem is so hard that the model fails to generate any correct solutions in a group (Pg = 0), the length reward mechanism (which is designed to encourage longer, exploratory paths for hard problems) will never be triggered. The model receives no signal to try harder, precisely when it needs it most.

- The authors themselves acknowledge that the method's effectiveness relies on sufficient length variation among the sampled responses. If the model's sampling process (governed by temperature and top-p) produces responses of very similar lengths for a given problem, the standard deviation (σi) will be close to zero, making the standardized length (zi) unstable and the resulting length reward signal meaningless. This makes the framework's success sensitive to the initial state of the model and the choice of decoding hyperparameters.

- The preliminary analysis (Figure 1) shows that pass@32 (a proxy for the training objective) generally improves with longer responses, while pass@1 (the primary evaluation metric) decreases. The framework is trained to find at least one correct answer within a group, which favors exploration. However, it is evaluated on its ability to produce a correct answer on the first attempt (pass@1), which favors exploitation and conciseness. While the paper shows success on both, this fundamental tension between the training signal and the desired inference behavior is a subtle but significant weakness.

- The main comparison is against DeepMath-Zero, a vanilla reinforcement learning approach. The ablation studies use fixed "Length Penalty" and "Length Bonus" strategies, which are arguably strawman arguments. The paper would be stronger if it compared DeepCompress against other adaptive length-reward strategies mentioned in its own related work section, which would provide a more rigorous test of its claimed superiority.

**Questions:**

See weaknesses.

---

> ### Author Response · Authors · 2025-11-24
> **Response to Reviewer kWZ8 (1 of 2)**
>
> We appreciate your thorough review and valuable feedback. We address the comments below:
>
> > W1. Concern about batch pass ratio
>
> DAPO proposes to over-sample and filter out prompts with the accuracy equal to 1 and 0. In fact, our sampling batch size are four times larger than training, reaching 2048. As shown in the table below, the batch pass ratio increases very steadily during the training process.
>
> | **Batch Pass Ratio** | 20   | 40   | 60   | 80   | 100  | 120  | 140  | 160  | 180  | 200  | 220  | 240  | 260  | 280  | 300  | 320  | 340  | 360  | 380  | 400  | 420  | 440  | 460  | 480  | 500  |
> | -------------------- | ---- | ---- | ---- | ---- | ---- | ---- | ---- | ---- | ---- | ---- | ---- | ---- | ---- | ---- | ---- | ---- | ---- | ---- | ---- | ---- | ---- | ---- | ---- | ---- | ---- |
> | DeepCompress-Zero-3B | 38.4 | 47.1 | 52.2 | 53.2 | 53.9 | 56.7 | 55.3 | 57.6 | 59.4 | 58.7 | 58.3 | 59.4 | 60.5 | 59.9 | 61.3 | 61.1 | 60.8 | 62.0 | 61.2 | 60.4 | 62.2 | 63.0 | 65.5 | 66.3 | 66.8 |
> | DeepCompress-Zero-7B | 55.8 | 61.7 | 64.9 | 66.6 | 68.6 | 69.7 | 70.0 | 70.9 | 71.7 | 73.5 | 75.8 | 76.2 | 77.0 | 76.6 | 77.8 | 77.1 | 77.4 | 77.6 | 79.3 | 79.0 | 79.3 | 79.8 | 80.6 | 80.9 | 80.9 |
>
> We recognize this is a common concern among reviewers. We have added a new figure (Figure 5 in Appendix C) plotting the complete training steps to address this.
>
> > W2. Correctness-conditioned length reward
>
> Thank you for your feedback, this is a very valuable suggestion. We attempted to apply the length reward to prompts where all rollouts were incorrect, and we found this leads to severe **reward hacking** due to the following two reasons: 1) DAPO standardizes the final advantage. 2) Since the outcome rewards are all zero, the advantage is directly determined by the length reward. This causes the model to reward long, incorrect responses as if they were correct answers. Consequently, while the model's response length sharply increased, there was no improvement in performance. Therefore, we followed the standard DAPO practice of filtering out completely incorrect examples to ensure training stability.
>
> > W3. Sensitive to the initial state or the decoding hyperparameters?
>
> We observed that when training on the Qwen-Math series models, the sampling diversity (in both length and answer format) is too low. Using such a model as the initial state, we found that not only our method fails, but many other length-reward RL methods also fail.
> **We want to clarify that this does not mean our method is sensitive.** This is a characteristic of RL: guiding and correcting based on the model's own behavior. If you want to completely change the model's behavior patterns, then SFT would be a better choice.

---

> ### Author Response · Authors · 2025-11-24
> **Response to Reviewer kWZ8 (2 of 2)**
>
> > W4. Explore during training and exploit during testing
>
> 1. We have added the pass@32 metric, and DeepCompress consistently achieves the best performance (especially on challenging benchmark like AIME 24/25).
>
> | Pass@32                  | MATH500 | AMC23 | Olympiad Bench | Minerva Math | AIME24 | AIME25 | Poly Math | Avg Acc |
> | ------------------------ | ------- | ----- | -------------- | ------------ | ------ | ------ | --------- | ------- |
> | DeepMath-Zero-3B         | 91.0    | 76.7  | 63.1           | 58.4         | 30.8   | 31.1   | 49.4      | 57.2    |
> | **DeepCompress-Zero-3B** | 91.5    | 76.7  | 63.5           | 57.9         | 33.7   | 32.5   | 52.4      | 58.3    |
> | DeepMath-Zero-7B         | 95.9    | 89.3  | 72.6           | 68.9         | 46.5   | 36.1   | 59.2      | 66.9    |
> | **DeepCompress-Zero-7B** | 96.1    | 88.7  | 72.4           | 66.4         | 53.1   | 40.4   | 58.1      | 67.9    |
>
> 2. This is precisely where the core novelty of our dynamic strategy lies. DeepCompress teaches the model through the following steps:
>
> ​	1）For hard problems, it encourages longer responses to reinforce the ability to explore correct answers, promoting generalization unsolvable problem sets.
>
> ​	2）Once a correct answer is sampled, the model receives a positive training signal that reinforces its mathematical reasoning capabilities.
>
> ​	3）As the model's capability improves and it successfully solves more problems, it learns to solve efficiently.
>
> Throughout the **Exploration-to-Exploitation** process, DeepCompress "recalls" problems that originally yielded no positive training signals. This explains why our model achieves a higher performance upper bound (pass@32).
>
> > W5. More length control baselines
>
> In addition to the existing Open-Reasoner-Zero and SimpleRL-Zoo baselines, we have compared against other length-penalty strategies such as those in Kimi-k1.5 [1] and Short-RL [2], with the results presented below.
>
> This finding is consistent with our preliminary experimental conclusions: while these methods achieve excellent results in length compression, their performance is simultaneously affected. DeepCompress, through its dynamic training strategy, aims to achieve the optimum performance in both efficiency and performance.
>
> | Pass@1               | MATH500  | AMC23    | Olympiad Bench | Minerva Math | AIME24   | AIME25  | Poly Math | Avg Acc  |
> | -------------------- | -------- | -------- | -------------- | ------------ | -------- | ------- | --------- | -------- |
> | Kimi-k1.5            | 53.4     | 36.6     | 19.7           | 20.3         | 6.7      | 1.2     | 28.2      | 23.7     |
> | Short-RL             | 63.4     | 35.2     | 26.1           | 31.1         | 8.1      | 1.9     | 29.4      | 27.9     |
> | DeepCompress-Zero-3B | **75.3** | **49.4** | **39.3**       | **32.7**     | **16.7** | **7.1** | **35.8**  | **36.6** |
>
> | Length               | MATH500 | AMC23 | Olympiad Bench | Minerva Math | AIME24 | AIME25 | Poly Math | Avg Len |
> | -------------------- | ------- | ----- | -------------- | ------------ | ------ | ------ | --------- | ------- |
> | Kimi-k1.5            | 162     | 102   | 274            | 182          | 70     | 222    | 80        | 156     |
> | Short-RL             | 411     | 513   | 752            | 251          | 843    | 445    | 342       | 508     |
> | DeepCompress-Zero-3B | 2073    | 3986  | 4138           | 2370         | 6808   | 4130   | 3382      | 3841    |
>
> [1] Kimi k1.5: Scaling Reinforcement Learning with LLMs
>
> [2] Efficient RL Training for Reasoning Models via Length-Aware Optimization

---

> > ### Comment · Reviewer_kWZ8 · 2025-11-27
> >
> > Thank you for the detailed response. Most of my concerns are addressed. I have updated the rating.

---

> > > ### Author Response · Authors · 2025-11-27
> > >
> > > Thank you for your valuable feedback and the updated rating. We are very glad that the additional results helped clarify the points you raised, and we are willing to continue the discussion if there are any further questions :)

---

### Author Response · Authors · 2025-12-03
**Summary of Rebuttal & Updates for the New AC (3 of 3)**

Dear reviewer,

We sincerely thank all reviewers for their constructive comments, which have provided a valuable opportunity to clarify our contributions and further substantiate the robustness of our work.

Our preliminary experiments reveal that consistently favoring shorter responses may inadvertently suppress the upper bound of reasoning potential (i.e., solution coverage). We argue that for challenging problems, the community and users prioritize correctness over inference cost—allocating a larger computational budget is worthwhile if it ensures the correct solution.

DeepCompress addresses this by establishing a paradigm where the model **autonomously** learns to compress reasoning for "simple" questions while actively exploring for "hard" ones. We hope this adaptive strategy provides valuable inspiration for the community to develop LRMs that are both efficient and powerful, without sacrificing the capability to solve complex problems.

Best,

Authors

---

### Author Response · Authors · 2025-12-03
**Summary of Rebuttal & Updates for the New AC (2 of 3)**

3. **Hyper-parameter Analysis (Addressing R-EuSL, R-GJ2N)** We performed comprehensive ablations on $\alpha$, $\beta$, and $\lambda$ to verify robustness:

- **Robustness ($\alpha$):** The model consistently outperforms the previous SOTA (34.6 from DeepMath-Zero-3B) across a wide range of reward weights.

    | alpha       | 0.1  | 0.2  | 0.5  | 1.0  |
    | ----------- | ---- | ---- | ---- | ---- |
    | Avg Acc | 36.1 | 36.6 | 36.7 | 35.9 |

- **Dynamic Mechanism ($\beta$):** We validated that our **Model-Aware Difficulty** (dynamic $\beta$) significantly outperforms static **Data-Labeled Difficulty** in both accuracy and efficiency.

    | Pass@1                        | MATH500  | AMC23    | Olympiad Bench | Minerva Math | AIME24   | AIME25  | Poly Math | Avg Acc  |
    | ----------------------------- | -------- | -------- | -------------- | ------------ | -------- | ------- | --------- | -------- |
    | Data-Labeled Difficulty       | 72.7     | 43.9     | 36.0           | 31.8         | 9.4      | 5.0     | 32.7      | 33.1     |
    | Model-Aware Difficulty (Ours) | **75.3** | **49.4** | **39.3**       | **32.7**     | **16.7** | **7.1** | **35.8**  | **36.6** |

    | Length                        | MATH500 | AMC23 | Olympiad Bench | Minerva Math | AIME24 | AIME25 | Poly Math | Avg Len |
    | ----------------------------- | ------- | ----- | -------------- | ------------ | ------ | ------ | --------- | ------- |
    | Data-Labeled Difficulty       | 2,566   | 5,407 | 5,563          | 1,833        | 10,297 | 6,628  | 5,145     | 5,348   |
    | Model-Aware Difficulty (Ours) | 2,073   | 3,986 | 4,138          | 2,370        | 6,808  | 4,130  | 3,382     | 3,841   |

- **Exploration ($\lambda$):** We found that a larger EMA factor ($\lambda=0.99$) is essential for encouraging early exploration.

    | lambda                   | 0.9  | 0.95 | 0.99 |
    | ------------------------ | ---- | ---- | ---- |
    | DeepCompress-Zero-3B | 35.5 | 36.5 | 36.6 |
    | DeepCompress-Zero-7B | 44.5 | 48.9 | 48.7 |

4. **Expanded Generalization Benchmarks (Addressing R-EuSL)** To validate that our method generalizes beyond mathematical reasoning, we added evaluations on **GPQA-Diamond** (biology, physics and chemistry), **Big Bench Hard** (BBH) (complex multi-step reasoning), and **MMLU-STEM** (science/engineering reasoning). DeepCompress serise models consistently outperforms all baselines.

    | Pass@1                   | GPQA     | BBH      | **MMLU-STEM** |
    | ------------------------ | -------- | -------- | ------------- |
    | Qwen-2.5-3B              | 21.0     | 7.5      | 48.1          |
    | Qwen-2.5-3B-Instruct     | 29.9     | 48.7     | 71.3          |
    | DeepMath-Zero-3B         | 30.2     | 54.7     | 71.6          |
    | **DeepCompress-Zero-3B** | **31.7** | **56.0** | **73.3**      |
    | Qwen-2.5-7B              | 25.3     | 12.1     | 41.3          |
    | Open-Reasoner-Zero-7B    | 38.1     | 47.0     | 83.2          |
    | Qwen-2.5-7B-SimpleRL-Zoo | 30.2     | 15.0     | 74.9          |
    | DeepMath-Zero-7B         | 42.6     | 72.7     | 85.0          |
    | **DeepCompress-Zero-7B** | **43.9** | **75.5** | **85.7**      |

---

### Author Response · Authors · 2025-12-03
**Summary of Rebuttal & Updates for the New AC (1 of 3)**

We sincerely thank the Area Chair and all reviewers for their time and constructive feedback. We are very glad that the additional results helped address the reviewers' concerns during the rebuttal. We deeply appreciate that **three reviewers have raised their scores (4$\to$6)**.

Regarding **Reviewer EuSL** (currently rating 6), although they have not yet had the chance to respond, we followed their suggestions to add expanded generalization benchmarks (GPQA, BBH, MMLU-STEM) and hyper-parameter analysis. We believe they will be **delighted to see** that our method remains robust and consistently outperforms baselines.

We summarize the key updates below:

1. **Verification of Training Stability (Addressing R-kWZ8, R-EuSL, R-GJ2N, R-t6ev)** We addressed concerns regarding the **Batch Pass Ratio** by clarifying that our large sampling batch size (2048) ensures a robust learning signal. We have added **Figure 5 in Appendix C** to provide further details, showing the pass ratio increases steadily throughout training (e.g., for DeepCompress-Zero-3B, rising from 38.4% to 66.8%), confirming the stability of our "Exploration-to-Exploitation" process.

    | **Batch Pass Ratio** | 20   | 40   | 60   | 80   | 100  | 120  | 140  | 160  | 180  | 200  | 220  | 240  | 260  | 280  | 300  | 320  | 340  | 360  | 380  | 400  | 420  | 440  | 460  | 480  | 500  |
    | -------------------- | ---- | ---- | ---- | ---- | ---- | ---- | ---- | ---- | ---- | ---- | ---- | ---- | ---- | ---- | ---- | ---- | ---- | ---- | ---- | ---- | ---- | ---- | ---- | ---- | ---- |
    | DeepCompress-Zero-3B | 38.4 | 47.1 | 52.2 | 53.2 | 53.9 | 56.7 | 55.3 | 57.6 | 59.4 | 58.7 | 58.3 | 59.4 | 60.5 | 59.9 | 61.3 | 61.1 | 60.8 | 62.0 | 61.2 | 60.4 | 62.2 | 63.0 | 65.5 | 66.3 | 66.8 |
    | DeepCompress-Zero-7B | 55.8 | 61.7 | 64.9 | 66.6 | 68.6 | 69.7 | 70.0 | 70.9 | 71.7 | 73.5 | 75.8 | 76.2 | 77.0 | 76.6 | 77.8 | 77.1 | 77.4 | 77.6 | 79.3 | 79.0 | 79.3 | 79.8 | 80.6 | 80.9 | 80.9 |

2. **Superiority over Length-Control Baselines (Addressing R-kWZ8, R-EuSL, R-t6ev)** A primary concern was the comparison with other length-penalty strategies. In addition to the existing Open-Reasoner-Zero and SimpleRL-Zoo baselines, We conducted experiments against **Kimi-k1.5** [1] and **Short-RL** [2]. The results demonstrate that while these baselines achieve length compression, they suffer significant performance degradation on reasoning tasks. In contrast, DeepCompress optimizes efficiency without compromising accuracy.

    | Pass@1               | MATH500 | AMC23 | Olympiad Bench | Minerva Math | AIME24 | AIME25 | Poly Math | Avg Acc |
    | -------------------- | ------- | ----- | -------------- | ------------ | ------ | ------ | --------- | ------- |
    | Kimi-k1.5            | 53.4    | 36.6  | 19.7           | 20.3         | 6.7    | 1.2    | 28.2      | 23.7    |
    | Short-RL             | 63.4    | 35.2  | 26.1           | 31.1         | 8.1    | 1.9    | 29.4      | 27.9    |
    | DeepMath-Zero-3B     | 72.8    | 48.0  | 38.0           | 30.8         | 11.5   | 6.9    | 34.1      | 34.6    |
    | DeepCompress-Zero-3B | 75.3    | 49.4  | 39.3           | 32.7         | 16.7   | 7.1    | 35.8      | 36.6    |

    | Length               | MATH500 | AMC23 | Olympiad Bench | Minerva Math | AIME24 | AIME25 | Poly Math | Avg Len |
    | -------------------- | ------- | ----- | -------------- | ------------ | ------ | ------ | --------- | ------- |
    | Kimi-k1.5            | 162     | 102   | 274            | 182          | 70     | 222    | 80        | 156     |
    | Short-RL             | 411     | 513   | 752            | 251          | 843    | 445    | 342       | 508     |
    | DeepMath-Zero-3B     | 5,441   | 6,097 | 8,496          | 14,128       | 10,922 | 10,524 | 8,385     | 9,142   |
    | DeepCompress-Zero-3B | 2,073   | 3,986 | 4,138          | 2,370        | 6,808  | 4,130  | 3,382     | 3,841   |

[1] Team, Kimi, et al. Kimi k1. 5: Scaling reinforcement learning with llms.

[2] Yuan, Danlong, et al. Efficient RL Training for Reasoning Models via Length-Aware Optimization.

---

### Meta-Review · Area_Chair_S72G · 2026-01-24

**Summary:**

DeepCompress proposes an adaptive RL reward that shortens reasoning for “easy” prompts and encourages longer exploration for “hard” prompts, where difficulty is defined online via group pass rate relative to the batch average. Reviewers agreed the problem is timely and the empirical results are strong: the method improves accuracy and token efficiency over prior “always penalize length” approaches and strong Zero-RL baselines. Most initial concerns were about missing baselines, ablations, and stability of the batch-relative difficulty signal; the rebuttal added substantial experiments that strengthen the case.

**Reviewer Concerns:**

Addressed:

* Missing baselines: Added comparisons to Kimi-k1.5 and Short-RL; results support that DeepCompress avoids the accuracy drop seen in strong length-penalty methods.

* Ablations / hyperparameters: Added sensitivity for α and λ, plus a comparison of model-aware vs. data-labeled difficulty.

* Generalization: Added GPQA, BBH, MMLU-STEM showing gains beyond math.

* Stability: Provided evidence that batch pass ratio trends are stable under their large-sample training recipe.

Partially addressed / still outstanding:

* Difficulty is relative to batch composition (Pg vs Pb), so labels can fluctuate under skewed batches.

* Correctness-conditioned length reward means no “explore longer” signal when all rollouts are wrong (authors explain reward hacking otherwise).

* Practical reliance on sufficient sampling diversity/length variance; sensitivity to base model + decoding is not fully characterized.

**Reviewer Scores:**

* kWZ8: 4 ->  likely 6 (they indicated most concerns resolved and updated).

* EuSL: 6  -> likely 6–7 given added baselines, ablations, and non-math evals.

* GJ2N: 4  -> 6 (explicitly updated).

* t6ev: 4  -> 6 (explicitly updated).

---

### Decision · Program_Chairs · 2026-01-26

Accept (Poster)